# 🥤 LATTE: Latent Trajectory Embedding for Diffusion-Generated Image Detection

## Abstract

The rapid advancement of diffusion-based image generators has made it increasingly difficult to distinguish generated from real images. This erodes trust in digital media, making it critical to develop generated image detectors that remain reliable across different generators. While recent approaches leverage diffusion denoising cues, they typically rely on single-step reconstruction errors and overlook the sequential nature of the denoising process. In this work, we propose **LATTE** - **Lat**ent **T**rajectory **E**mbedding - a novel approach that models the evolution of latent embeddings across multiple denoising steps. Instead of treating each denoising step in isolation, LATTE captures the trajectory of these representations, revealing subtle and discriminative patterns that distinguish real from generated images. Experiments on several benchmarks, such as GenImage, Chameleon, and Diffusion Forensics, show that LATTE achieves superior performance, especially in challenging cross-generator and cross-dataset scenarios, highlighting the potential of latent trajectory modeling.

## 1 Introduction

Diffusion-based generative models have fundamentally transformed the field of image generation (Ho et al., 2020; Song et al., 2020; Rombach et al., 2022a; Nichol et al., 2021; Dhariwal & Nichol, 2021; Saharia et al., 2022; Podell et al., 2023; Midjourney, 2024; Black Forest Labs, 2025). These models generate photorealistic content - such as portraits, landscapes, and complex scenes - by iteratively adding and then removing noise from data or latent representations, typically guided by a text prompt (Rombach et al., 2022b). While this progress has unlocked transformative and creative applications, it has also facilitated the creation of fake images that are hard to visually distinguish from authentic content. Such capabilities have already been exploited by malicious actors, for instance, to create fraudulent impersonations of public figures (Twomey et al., 2023; de Rancourt-Raymond & Smaili, 2023) or fabricate "evidence" in legal disputes (Delfino, 2022; Sandoval et al., 2024; Koutras & Selvadurai, 2024). The challenge is also amplified by the growing landscape of image generation models, each introducing its own artifacts and characteristics. This underscores the urgent need for robust detectors able to distinguish real from generated images.

Recent efforts to detect generated images (Wang et al., 2023; Zhang & Xu, 2023; Ma et al., 2023; Luo et al., 2024; Ricker et al., 2024; Chen et al., 2024; Chu et al., 2024; Yan et al., 2025; Cheng et al., 2025) leverage distinctive signatures left by the generative process. Based on the hypothesis that diffusion models can reconstruct synthetic images more accurately than real ones, methods like DIRE (Wang et al., 2023) and LaRE (Luo et al., 2024) define novel representations that capture the error between an input image and its reconstruction. While achieving solid performance, these approaches rely on single-step representations and overlook the inherent sequential nature of denoising - a process that largely underlies the synthetic artifacts of fake images. We address this by treating the sequence of latent representations as a distinctive signature.

In this paper, we introduce **Lat**ent **T**rajectory **E**mbedding - **LATTE**, a novel approach that explicitly models the evolution of latent representations across multiple denoising steps. Namely, diffusion models generate images through a sequence of gradual denoising steps, where each learned update iteratively refines the sample toward the data manifold. This iterative process defines a trajectory

that reflects how the model interprets and refines the underlying content. We hypothesize that real images, whose details and textures can lie outside the model's learned manifold, will often produce small inconsistencies between successive denoising steps. On the contrary, fake images will follow smoother, more self-consistent trajectories aligned with the model's generative prior. Specifically, given an image, we leverage a pretrained latent diffusion model to obtain its latent embedding. We apply standard forward noising and then extract intermediate latent states during the denoising at evenly spaced steps. This spacing provides a representative view of early, middle, and late denoising stages, capturing the full spectrum of the denoising dynamics. The resulting trajectory reflects how the internal representation evolves across steps, but it does not reveal which image regions drive these changes. To enrich the trajectory signal, we fuse each latent with visual features extracted from a pretrained image encoder using a stack of transformer decoders. The enriched sequence is subsequently aggregated into a compact representation, combined with global image features, and passed to a lightweight classifier. This combination of latent dynamics and semantic cues enables LATTE to leverage subtle inconsistencies indicative of generated content.

We evaluate LATTE on well-established benchmarks for generated image detection, namely GenImage (Zhu et al., 2023b), Chameleon (Yan et al., 2025), and Diffusion Forensics (Wang et al., 2023). Our model surpasses current state-of-the-art methods, achieving an average improvement of 4.1% on GenImage over AIDE (Yan et al., 2025) and 7.1% in cross-domain settings on Diffusion Forensics over LaRE (Luo et al., 2024). In particular, on one of the most challenging subsets of GenImage i.e., BigGAN (Brock et al., 2018), LATTE outperforms the most competitive baseline by 9.5%, highlighting its cross-generator generalizability. In cross-domain settings - for instance, the Bedroom partition of Diffusion Forensics - we observe a 11.1% gain, underscoring LATTE's robustness to specialized domains.

In summary, our contributions are threefold: (1) We propose LATTE, the first diffusion-based embedding that explicitly leverages the trajectory of latent states across multiple denoising steps. (2) We introduce a two-stage architecture that (i) samples and enriches latent trajectories via transformer decoders and (ii) aggregates the latent embeddings into a compact and discriminative representation. (3) We demonstrate that LATTE achieves state-of-the-art performance and exhibits strong performance across diverse benchmarks, unseen generators, perturbations, and domains.

## 2 RELATED WORK

**Image Generation Models.** Early methods for image generation were predominantly based on Generative Adversarial Networks (GANs) (Goodfellow et al., 2020; Karras et al., 2017; Brock et al., 2018; Choi et al., 2018; Park et al., 2019; Zhu et al., 2017), Variational Autoencoders (VAEs) (Kingma et al., 2013; Sohn et al., 2015; Zhao et al., 2017; Van Den Oord et al., 2017), and autoregressive models (Van den Oord et al., 2016; Parmar et al., 2018; Esser et al., 2021; Ramesh et al., 2021). GANs produce realistic images, but are hard to train and lack stable likelihood estimation. VAEs enable efficient inference and structured latent spaces but tend to generate blurry images. Autoregressive models offer precise likelihood modeling but suffer from slow, sequential sampling, especially at high resolutions.

To address the limitations of early methods, Denoising Diffusion Probabilistic Models (DDPMs) (Ho et al., 2020) introduced a generative process that reverses a gradual noising procedure, offering stable likelihood-based training and state-of-the-art image quality. Further advancements have explored improved sampling efficiency (Song et al., 2020), accelerated solvers (Karras et al., 2022), architectural refinements (Saharia et al., 2022; Nichol et al., 2021), and improved conditional generation with classifier-free guidance (Ho & Salimans, 2022). Latent Diffusion Models (LDMs) (Rombach et al., 2022a) improved scalability by operating in a compressed latent space learned via a variational autoencoder, enabling high-resolution generation at much lower cost. LDMs underpin popular models like Stable Diffusion, and have enabled extensions such as ControlNet (Zhang et al., 2023) for spatial conditioning, SDXL (Podell et al., 2023) for ultra-high-resolution output, and LCM (Luo et al., 2023) for efficient few-step sampling. Diffusion models now represent the primary focus of current research in generated image detection, as also addressed in this paper.

**Detection of Generated Images.** Early efforts in generated image detection targeted GAN-generated content, starting with handcrafted features (Yang et al., 2019; Liy & InIctuOculi, 2018) and later advancing to convolutional neural networks (CNNs) trained on datasets like FaceForensics++ (Rossler

et al., 2019). Subsequent works investigated intrinsic manipulation traces such as spectral artifacts in the frequency domain (Luo et al., 2021; Frank et al., 2020) and inconsistencies in noise distributions (Wang & Chow, 2023; Bai et al., 2024). While these approaches improved robustness across GANs, they demonstrated limited generalization to diffusion-generated images.

To improve the generalizability of methods for detecting diffusion-based images, recent work has explored strategies that leverage the internal mechanics of the diffusion process. Some approaches focus on full image reconstruction: DIRE (Wang et al., 2023) introduced the idea of using DDIM (Song et al., 2020) inversion error as a discriminative feature, while DRCT (Chen et al., 2024) uses a contrastive training objective on reconstructed images. Other methods, like LaRE (Luo et al., 2024), improve efficiency by operating in latent space and using a single-step inversion. AIDE (Yan et al., 2025) incorporates low-level patch statistics and high-level semantics. In contrast, our method leverages the trajectory of latent states across denoising steps, capturing the evolution of the process as a more discriminative representation.

Another line of research explores powerful vision encoders, such as CLIP (Radford et al., 2021), used either as a frozen feature extractor with downstream classifiers (Zhang et al., 2024), or in fine-tuned multi-modal frameworks aligning image and text embeddings to capture inconsistencies in generated content (Cozzolino et al., 2024; Li et al., 2024). We also employ CLIP's vision encoder, alongside other large-scale vision encoders, to enrich our proposed latent trajectory embedding.

## 3 METHODOLOGY

In this section, we introduce our **Lat**ent **T**rajectory **E**mbedding (**LATTE**) for generated image detection. First, we give a brief overview of the denoising process in latent diffusion models. Then, we continue by introducing LATTE and explaining how to extract and fuse a sequence of latents with visual features. Finally, we show how LATTE can be aggregated into a unified representation to enhance the detection of generated images.

### 3.1 PRELIMINARIES

**Diffusion Probabilistic Models.** Diffusion models define a Markov chain of diffusion steps that progressively add Gaussian noise to data until turning it into noise. In the literature, this is referred to as a forward noising process (Ho et al., 2020). Specifically, starting from a clean image $x$, the forward chain gradually injects Gaussian noise over $T$ discrete steps:

$$q(x_t | x_{t-1}) = \mathcal{N}\big(x_t; \sqrt{\alpha_t}\, x_{t-1},\, (1-\alpha_t)\mathbf{I}\big), \tag{1}$$

where $x_t$ is the noisy image at step $t$ and the schedule $\{\alpha_t\}$ controls the noise variance at each step. After $T$ steps, the image becomes nearly isotropic noise. In the reverse process, also defined as a Markov chain, the noisy image is gradually denoised to obtain the raw image. This backward chain leverages a neural network $\epsilon_\theta(x_t, t)$ parameterized by $\theta$ to predict and remove this noise, defined as:

$$p_\theta(x_{t-1} | x_t) = \mathcal{N}\big(x_{t-1};\, \tfrac{1}{\sqrt{\alpha_t}}\big(x_t - (1-\alpha_t)\epsilon_\theta(x_t, t)\big),\, \sigma_t^2 \mathbf{I}\big). \tag{2}$$

**Latent Diffusion.** To improve efficiency, latent diffusion models (Rombach et al., 2022a) first encode images into a lower-dimensional latent space via a pretrained VAE encoder $E_{\text{VAE}}$, producing $z_0 = E_{\text{VAE}}(x_0)$. The forward and reverse processes then operate on these latent codes $z_t \in \mathbb{R}^d$:

$$q(z_t | z_{t-1}) = \mathcal{N}\big(z_t; \sqrt{\alpha_t}\, z_{t-1},\, (1-\alpha_t)\mathbf{I}\big), \tag{3}$$

$$p_\theta(z_{t-1} | z_t) = \mathcal{N}\big(z_{t-1};\, \mu_\theta(z_t, t),\, \Sigma_\theta(z_t, t)\big). \tag{4}$$

After denoising to $z_0$, a VAE decoder $D_{\text{VAE}}$ reconstructs the final image $\hat{x} = D_{\text{VAE}}(z_0)$. Latent diffusion thus preserves high sample quality while reducing computational and memory demands.

### 3.2 LATTE: LATENT TRAJECTORY EMBEDDING

In diffusion models, an image is reconstructed from noise by iteratively denoising latent variables over a sequence of timesteps (see Eqs. (3)–(4)). LATTE leverages the sequential structure of diffusion models by explicitly modeling how the latent embedding evolves across denoising steps. Instead of

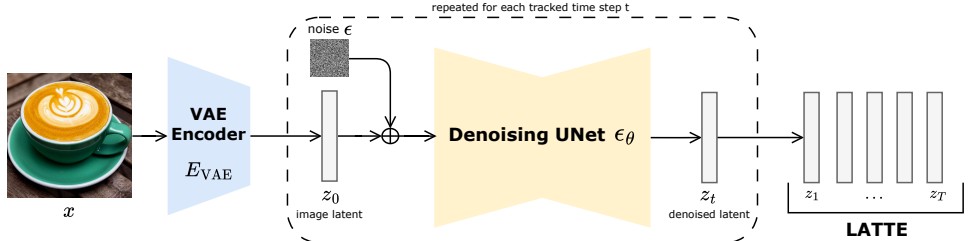

Figure 1: **Extraction of LATTE representation.** We construct the LATTE sequence by performing a single-step reconstruction for a selection of timesteps throughout the whole trajectory.

performing the full reverse chain, which is computationally expensive, we approximate intermediate states using single-step denoising at selected timesteps.

Given an input image $x$, we first encode it into latent space using a pretrained VAE encoder: $z_0 = E_{\text{VAE}}(x)$, as explained in section 3.1. Next, for each selected timestep $t$, we simulate the forward diffusion process in one closed-form operation, producing a noisy latent:

$$z_t = \sqrt{\bar{\alpha}_t}\, z_0 + \sqrt{1 - \bar{\alpha}_t}\, \epsilon, \quad \epsilon \sim \mathcal{N}(0, I), \tag{5}$$

where $\bar{\alpha}_t = \prod_{s=1}^{T} \alpha_s$ accumulates the noise schedule up to $t$. We then approximate the reverse diffusion at $t$ by performing a single denoising update using the pretrained UNet's noise predictor $\epsilon_\theta$:

$$\hat{z}_t = z_t - \sqrt{1 - \alpha_t}\, \epsilon_\theta(z_t, t). \tag{6}$$

This one-step correction yields an estimate $\hat{z}_t$ that closely approximates the latent at $t$, while avoiding the overhead of a full reverse pass from $T$ to $t$. By repeating this forward-then-single-step reverse procedure for each of the $T$ timesteps $\{t, \ldots, T\}$ chosen to uniformly span the denoising schedule, we assemble the latent trajectory embeddings: $\mathcal{T}(x) = \{\hat{z}_1, \hat{z}_2, \ldots, \hat{z}_T\}$, illustrated in Figure 1.

### 3.3 ARCHITECTURE DETAILS

Our architecture grounds the latent trajectory in visual context, ensuring that the latent representations remain tied to the image content. As illustrated in Figure 2, it unifies two complementary feature streams - the LATTE sequence and visual semantics - through two main stages: *Latent–Visual Fusion* and *Latent-Visual Classifier*.

**Latent–Visual Fusion.** Each latent embedding is enhanced through cross-attention with spatial features extracted from a pretrained vision encoder, to ground the denoising trajectory in the image content. Given an input image $x$, the vision encoder produces two outputs: (1) patch-level visual embeddings $V \in \mathbb{R}^{N \times d}$, and (2) a global image token $\mathbf{v}_{\text{IMG}} \in \mathbb{R}^d$. The patch tokens V capture fine-grained spatial and semantic information from the image and are leveraged for the refinement of the LATTE representation. The $\mathbf{v}_{\text{IMG}}$ token provides a holistic representation of the image and is used in the second stage.

Each latent embedding $\hat{z}_t$ in the trajectory $\mathcal{T}(x)$ is first flattened and linearly projected to match the dimensions $d$ of the visual features V. The projected latents are then independently enhanced using a stack of $L$ transformer decoders, each consisting of a cross-attention layer followed by a feed-forward layer, with residual connections and layer normalization. Specifically, each latent $\hat{z}_t$ attends to the patch-level visual embeddings V using multi-head cross-attention (MHA) mechanism:

$$\tilde{z}_t = \text{MHA}(Q, K, V) = [head_1, \ldots head_h]\mathbf{W}^O, \text{where } head_i = \text{softmax}\left(\frac{\hat{z}_t K^\top}{\sqrt{d}}\right) V, \tag{7}$$

where the keys and values $K, V \in \mathbb{R}^{N \times d}$ are both the visual features V, $\mathbf{W}^O$ is the output projection layer, $h$ is the number of heads and $d$ is the dimension of the embeddings. Each $\hat{z}_t$ is processed through $L$ such layers, allowing it to align with different spatial features in the image independently of the other timesteps.

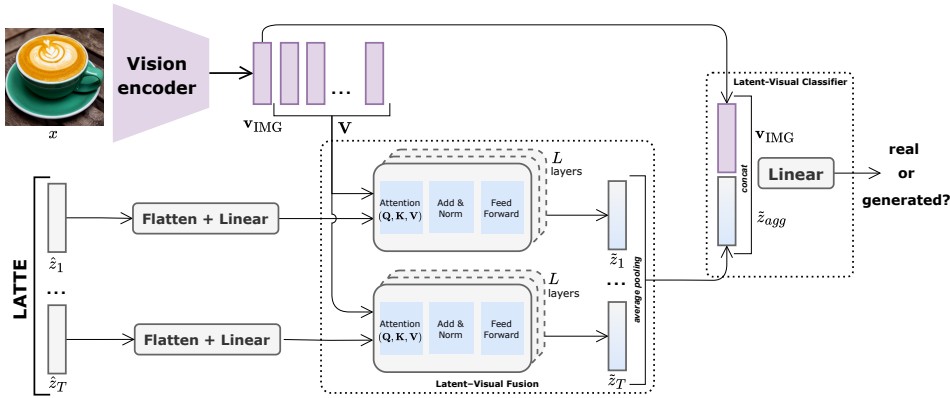

Figure 2: **Overview of our proposed architecture using LATTE.** It encompasses two stages: (1) *Latent–Visual Fusion*, where the LATTE is fused with visual semantics through stacks of $L$ cross-attention layers, and (2) *Latent-Visual Classifier* for average aggregation and output prediction.

**Latent-Visual Classifier.** After enhancing each latent embedding through $L$ transformer decoder layers, we obtain a set of enriched embeddings $\tilde{z}_1, \ldots, \tilde{z}_T$. To aggregate this sequence into a unified representation, we perform average pooling across all $T$ latents: $\tilde{z}_{\text{agg}} = \frac{1}{T} \sum_{i=1}^{T} \tilde{z}_t$. Alternatively, we can perform $CLS$ token pooling where a special token $z_{\text{CLS}}$ is prepended to the sequence of latents $z_{\text{CLS}}, \tilde{z}_1, \ldots, \tilde{z}_T$, processed with self-attention layers and then used as an aggregation $\tilde{z}_{\text{agg}}$. The aggregated representation encodes how the latent embeddings transition through successive denoising steps, effectively encoding the reconstruction trajectory. Next, to incorporate a holistic semantic-level context, we concatenate $\tilde{z}_{\text{agg}}$ with the global image token $\mathbf{v}_{\text{IMG}}$: $\mathbf{z} = [\tilde{z}_{\text{agg}} \parallel \mathbf{v}_{\text{IMG}}] \in \mathbb{R}^{2d}$.

Finally, we feed this joint embedding $\mathbf{z}$ into a lightweight linear classifier, which leverages this combined information to make a real-vs-generated prediction. By pooling the latent embeddings and grounding them in image semantics, our aggregation strategy effectively amplifies subtle artifacts that single-step or pixel-based methods tend to overlook.

## 4 EXPERIMENTS & RESULTS

### 4.1 EXPERIMENTAL SETUP

**Datasets.** We evaluate LATTE across several complementary settings. First, to assess overall detection and cross-generator robustness, we use the GenImage dataset[1] (Zhu et al., 2023b), which contains real and fake images generated by eight generative models, including diffusion- and GAN-based approaches. Next, to test performance under more visually-challenging scenarios, we use the Chameleon dataset (Yan et al., 2025), which includes high-quality synthetic images designed to reduce detection artifacts. To evaluate cross-domain generalization, we use the Diffusion Forensics dataset (Wang et al., 2023), which spans multiple visual domains such as bedrooms, churches, and faces. All images are resized to $224 \times 224$ before being passed to the model.

**Training & Evaluation.** We extract the latent trajectories using Stable Diffusion 2.1. We empirically chose five timesteps: [981, 741, 521, 261, 1] for extracting the latents, which evenly spread across the trajectory. The visual features are obtained with a pretrained ConvNeXt (Liu et al., 2022), yielding a dimension size of $512$. All models are trained by minimizing binary cross-entropy loss to convergence, monitored on a held-out validation split matching the training generator. We use a batch size of 32, AdamW (Loshchilov & Hutter, 2017) optimizer (lr = 1e-4, weight decay = 4e-5), and a cosine-annealed learning rate scheduler. The experiments are conducted on a single H100 GPU, by training for approximately 2 hours for a single epoch. To provide a comprehensive evaluation, we follow standard practice in detection tasks and evaluate our models using accuracy and average precision. The code repository, training, and evaluation details will be released.

---

[1] Licensed under CC BY-NC-SA 4.0.

## 4.2 COMPARISON TO BASELINES

We first evaluate our method on GenImage (Zhu et al., 2023b), which essentially tests cross-generator generalization. All models are trained on images generated by SDv1.4 and evaluated across eight different generators, with baseline results cited from Yan et al. (2025). The results, shown in Table 1, indicate that LATTE/Avg (using average pooling) achieves the highest average accuracy among a broad set of related methods, improving by 4.1% over the recent AIDE model, followed by LATTE/CLS (using CLS token pooling). Notably, on one of the most challenging subsets - BigGAN, LATTE/Avg surpasses the strongest prior baseline (Ojha et al., 2023) by 9.5%. Note that we continue using LATTE/Avg in the subsequent experiments as our best model, denoted as LATTE for brevity.

Table 1: **Comparison of LATTE to baselines on GenImage benchmark (Zhu et al., 2023b).** All methods are trained on SDv1.4 of GenImage and evaluated over eight image generators. LATTE/Avg achieves the best average accuracy, improving by 4.1% over state-of-the-art methods.

| Method | Midjourney | SD v1.4 | SD v1.5 | ADM | GLIDE | Wukong | VQDM | BigGAN | *Avg.* |
|---|---|---|---|---|---|---|---|---|---|
| CNNSpot (Wang et al., 2020) | 52.8 | 96.3 | 95.9 | 50.1 | 39.8 | 78.6 | 53.4 | 46.8 | 64.2 |
| F3Net (Qian et al., 2020) | 50.1 | 99.9 | 99.9 | 49.9 | 50.0 | 99.9 | 49.9 | 49.9 | 68.7 |
| Spec (Zhang et al., 2019) | 52.0 | 99.4 | 99.2 | 49.7 | 49.8 | 94.8 | 55.6 | 49.8 | 68.8 |
| GramNet (Liu et al., 2020) | 54.2 | 99.2 | 99.1 | 50.3 | 54.6 | 98.9 | 50.8 | 51.7 | 69.9 |
| DeiT-S (Touvron et al., 2021) | 55.6 | 99.9 | 99.8 | 49.8 | 58.1 | 98.9 | 56.9 | 53.5 | 71.6 |
| ResNet-50 (He et al., 2016) | 54.9 | 99.9 | 99.7 | 53.5 | 61.9 | 98.2 | 56.6 | 52.0 | 72.1 |
| DIRE (Wang et al., 2023) | 65.8 | 99.7 | 99.7 | 54.5 | 58.1 | 99.4 | 54.3 | 49.8 | 72.7 |
| UnivFD (Ojha et al., 2023) | 73.2 | 84.2 | 84.0 | 55.2 | 76.9 | 75.6 | 56.9 | 80.3 | 73.3 |
| Swin-T (Liu et al., 2021) | 62.1 | 99.9 | 99.8 | 49.8 | 67.6 | 99.1 | 62.3 | 57.6 | 74.8 |
| GenDet (Zhu et al., 2023a) | 89.6 | 96.1 | 96.1 | 58.0 | 78.4 | 92.8 | 66.5 | 75.0 | 81.6 |
| DRCT (Chen et al., 2024) | **94.6** | 99.8 | 99.8 | 61.8 | 65.9 | 99.9 | 74.8 | 58.8 | 82.1 |
| PatchCraft (Zhong et al., 2023) | 79.0 | 89.5 | 89.3 | 77.3 | 78.4 | 89.3 | 83.7 | 72.4 | 82.3 |
| Co-Spy (Cheng et al., 2025) | 83.4 | 96.8 | 96.7 | 67.2 | 93.0 | 95.9 | 78.8 | 65.2 | 84.6 |
| LaRE (Luo et al., 2024) | 74.0 | 100 | 99.9 | 61.7 | 88.5 | **100** | **97.2** | 68.7 | 86.2 |
| AIDE (Yan et al., 2025) | 79.4 | 99.7 | 99.8 | 78.5 | 91.8 | 98.7 | 80.3 | 66.9 | 86.9 |
| **LATTE**/CLS | 85.8 | 99.0 | 99.0 | **83.2** | 86.7 | 96.8 | 88.7 | 77.8 | 89.6 |
| **LATTE**/Avg | 88.8 | **100** | **99.9** | 74.0 | **95.8** | 98.9 | 80.8 | **89.8** | **91.0** |

Next, we train our model on each generator-specific subset of GenImage and test it across all other subsets. Figure 3 reports the averaged accuracies, where our model again achieves the best overall performance. These findings demonstrate that explicitly modeling the trajectory evolution in latent space yields stronger robustness and more reliable detection across diverse image generators.

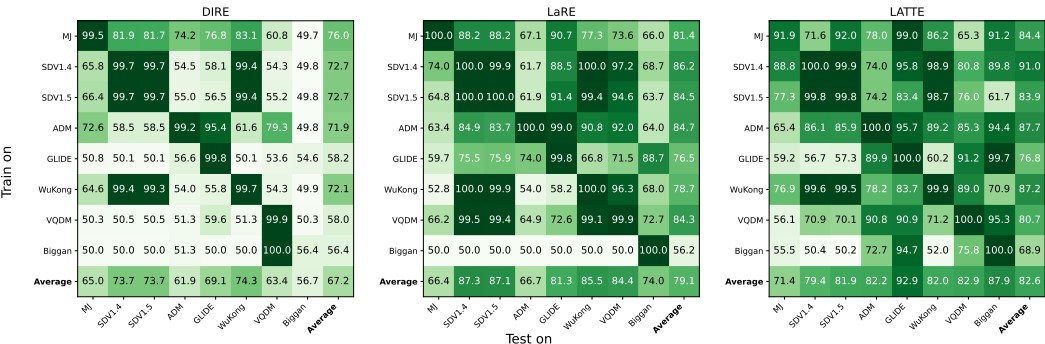

Figure 3: **Comparison of LATTE to baselines, by training and testing across all 8 generators of GenImage.** Each plot corresponds to one detector - DIRE (left; baseline), LaRE (center; baseline), and LATTE (right; proposed) - and shows the accuracy(%) when training on the subset listed on the vertical axis and testing on the subset listed along the horizontal axis.

We further evaluate our model on Chameleon (Yan et al., 2025), a recently proposed benchmark designed to reflect real-world scenarios by covering a broad range of content, including humans, animals, objects, and scenes. This benchmark allows us to test how well the model generalizes beyond its training distribution and captures transferable representations. As shown in Table 2, our method achieves consistent improvements over the baselines, achieving 2.5% improvement over AIDE when trained on the GenImage dataset. The results highlight both the robustness of our approach and its effectiveness in generalizing across diverse visual domains.

Table 2: **Cross-domain comparison on Chameleon (Yan et al., 2025).** Each column represents the accuracy (%) of different detectors, and the rows indicate the used training set.

| Training set | FreDect | LNP | UnivFD | DIRE | NPR | PatchCraft | CNNSpot | GramNet | AIDE | LATTE |
|---|---|---|---|---|---|---|---|---|---|---|
| SDv1.4 | 56.9 | 58.5 | 55.6 | 59.7 | 58.1 | 56.3 | 60.1 | 60.9 | 62.6 | **63.8** |
| All GenImage | 57.2 | 58.5 | 60.4 | 57.8 | 57.8 | 55.7 | 60.9 | 59.8 | 65.8 | **68.3** |

Finally, we evaluate the cross-domain generalization ability of LATTE on the Diffusion Forensics dataset (Wang et al., 2023). Specifically, we train LATTE on the SDv1.4 subset of GenImage and use LaRE and AIDE trained on the same data for a fair comparison. Table 3 reports accuracies across various generators and three distinct domains - Bedroom, ImageNet, and CelebA - which differ substantially from GenImage in both content and style. Across all three domains, LATTE consistently outperforms both LaRE and AIDE, achieving improvements such as 11.1% on Bedroom and 4% on Imagenet, with an overall average gain of 7.1%. We also notice that in-domain performance (train and test on the same data) is already saturated in prior work - often reported at or near 100% - so it offers limited insight into generalization. Therefore, we prioritized the evaluation in cross-generator and cross-domain settings.

Table 3: **Cross-domain comparison on Diffusion Forensics (Wang et al., 2023)**. LATTE achieves an overall average improvement of 7.1% accuracy over LaRE and 14.8% over AIDE.

(a) **Bedroom**

| Method | ADM | Dalle2 | DDPM | ProjGAN | StyleGAN | IDDPM | IF | LDM | MidJ | SDV | PNDM | VQDM | *Avg.* |
|---|---|---|---|---|---|---|---|---|---|---|---|---|---|
| LaRE | 53.0 | 66.7 | 57.6 | 50.5 | 62.3 | 55.2 | 90.9 | 93.5 | 90.9 | 78.4 | 53.5 | 82.4 | 69.5 |
| AIDE | 66.5 | **77.9** | **66.5** | 55.5 | 76.0 | 57.0 | 94.6 | 79.0 | **94.9** | 88.4 | 54.7 | 84.2 | 74.6 |
| **LATTE** | **89.9** | 76.5 | 63.3 | **65.3** | **93.5** | **90.0** | **99.9** | **99.3** | 91.2 | **91.9** | **75.3** | **92.6** | **85.7** |

(b) **ImageNet**

| Method | ADM | SDV | *Avg.* |
|---|---|---|---|
| LaRE | 81.4 | 98.5 | 89.9 |
| AIDE | 53.6 | **98.9** | 76.2 |
| **LATTE** | **89.8** | 98.0 | **93.9** |

(c) **Celeba**

| Method | Dalle2 | IF | MidJ | SDV | *Avg.* |
|---|---|---|---|---|---|
| LaRE | **77.7** | 96.3 | **90.9** | 95.2 | 90.0 |
| AIDE | 70.8 | 76.5 | 69.5 | 85.2 | 75.5 |
| **LATTE** | 77.5 | **96.7** | **90.9** | **99.6** | **91.1** |

## 4.3 ABLATION STUDY

In this section, we present ablation studies to quantify the contribution of each component, the impact of the denoising steps and vision backbone. Additional ablations are available in the Appendix A.

**Importance of each component.** We conduct an ablation study on three components: the visual features from the backbone, the latent trajectory from intermediate diffusion steps, and the Latent–Visual Fusion module that aligns these modalities via cross-attention. Four model variants are evaluated: (A) visual features only, (B) latent trajectory only, (C) visual + latent trajectory without fusion, and (D) the full model with all components. As shown in Table 4, both visual-only (A) and latent-only (B) variants perform poorly, confirming that neither modality alone is sufficient. Combining the two in (C) improves performance, indicating complementary cues, but the gains remain limited. The full model (D) achieves the best results across nearly all subsets, with large improvements on challenging cases such as VQDM (+11.6%) and BigGAN (+13.9%), underscoring the importance of effectively fusing latent and visual information.

Table 4: **Ablation on visual and latent components.** ✓ indicates that the component is included. Results are shown as Accuracy (%) for each generator. Including all components of our approach outperforms the visual-only and latent-only configurations by 16.1% and 37.8%.

| Model | Visual | Latent | Fusion | Midjourney | SDV1.4 | SDV1.5 | ADM | GLIDE | Wukong | VQDM | BigGAN | *Avg.* |
|---|---|---|---|---|---|---|---|---|---|---|---|---|
| A | ✓ | ✗ | ✗ | 83.5 | 99.9 | 99.9 | 51.7 | 56.2 | **99.9** | 58.4 | 50.0 | 74.9 |
| B | ✗ | ✓ | ✗ | 50.0 | 58.3 | 58.4 | 50.0 | 50.0 | 56.6 | 52.6 | 50.0 | 53.2 |
| C | ✓ | ✓ | ✗ | 80.5 | *100* | 99.9 | **76.7** | 84.3 | 99.8 | 69.2 | 75.8 | 85.7 |
| D | ✓ | ✓ | ✓ | **88.7** | **100** | 99.9 | 74.0 | **95.7** | 98.9 | **80.8** | **89.7** | **91.0** |

**Influence of denoising steps.** We study how performance changes with the number of denoising steps, varying $n \in \{1, 3, 5, 9, 13, 15\}$ used to sample intermediate latents for the trajectory. For

Table 5: **Accuracy(%) comparison of varying lengths of latent trajectory.** We compare the effect of different timestep configurations on the average accuracy across eight generative models. The best accuracy is achieved with the 5-timestep configuration ($n = 5$).

| $n$-steps | Midjourney | SDv1.4 | SDv1.5 | ADM | GLIDE | Wukong | VQDM | BigGAN | *Avg.* |
|---|---|---|---|---|---|---|---|---|---|
| 1 | 80.4 | 99.2 | 98.8 | 71.2 | 85.6 | 96.9 | 74.2 | 62.5 | 83.6 |
| 3 | 78.7 | 99.3 | 99.0 | **75.3** | 81.8 | 97.2 | 80.4 | 68.0 | 85.0 |
| **5** | **88.8** | **100** | **99.9** | 74.0 | **95.8** | **98.9** | **80.8** | **89.8** | **91.0** |
| 9 | 86.2 | 99.5 | 99.4 | 75.2 | 94.4 | 98.0 | 79.8 | 86.9 | 89.9 |
| 13 | 77.3 | 99.7 | 99.4 | 72.2 | 82.8 | 98.5 | 77.3 | 69.4 | 84.6 |
| 15 | 77.3 | 99.6 | 99.3 | 73.5 | 82.9 | 98.4 | 78.3 | 62.7 | 84.0 |

$n = 5$ steps, we empirically select the following: [981, 741, 521, 261, 1], while $n = 1$ corresponds to the single midpoint t = 521. The remaining configurations include both endpoints ($t = 1$ and $t \approx 1000$) with additional steps interpolated evenly across the trajectory. Our choice of such evenly spaced steps - spanning from near the start to the end of the trajectory - aims to capture the full spectrum of denoising behavior. As shown in Table 5, accuracy improves as the number of sampled steps increases, peaking at $n = 5$. Beyond this point, the $n = 9$ configuration maintains competitive results, but performance declines at 13 and 15 steps, suggesting that adding more steps introduces redundancy rather than additional useful information.

**Influence of vision backbone.** In our preliminary experiments, we used CLIP encoders (RN50, ViT-B/32), which underperformed on GenImage. This prompted the shift to other backbones: ConvNeXt-Base (Liu et al., 2022) pretrained on ImageNet-22k and CLIP ViT-L/14 (Ilharco et al., 2021), also leveraged by Ojha et al. (2023). Both improved the results significantly, with ConvNeXt consistently achieving the highest accuracy, as demonstrated in Table 6.

Table 6: **Accuracy(%) comparison between different vision backbones.** ConvNeXt outperforms CLIP ViT-L/14 by 5.3%.

| Vision encoder | Midjourney | SDv1.4 | SDv1.5 | ADM | GLIDE | Wukong | VQDM | BigGAN | *Avg.* |
|---|---|---|---|---|---|---|---|---|---|
| CLIP ViT-L/14 | **98.2** | *100* | **100** | 68.4 | 91.8 | **99.9** | 58.9 | 68.4 | 85.7 |
| **ConvNeXt** | 88.8 | **100** | 99.9 | **74.0** | **95.8** | 98.9 | **80.8** | **89.8** | **91.0** |

## 4.4 EMBEDDING SPACE ANALYSIS

We assess our model's discriminative capacity by visualizing real and generated image embeddings with t-SNE (van der Maaten & Hinton, 2008) in Figure 4. The first row depicts the embeddings extracted from the original, frozen ConvNeXt backbone, while the second row displays embeddings after the backbone has been fine-tuned with LATTE. The embeddings in the second row exhibit much clearer separation between real (blue) and generated (orange) samples, indicating reduced overlap and stronger class separation.

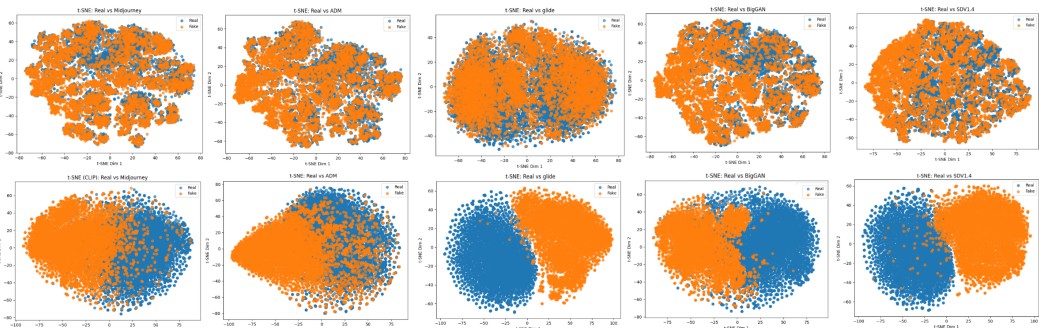

Figure 4: **Visualizations of t-SNE embeddings for real and fake images across five generators from GenImage.** The first row presents embeddings before using LATTE (extracted using the ConvNeXt), while the second row shows embeddings derived from LATTE. The much clearer separation in the second row illustrates LATTE's discriminative power.

## 4.5 ROBUSTNESS TO UNSEEN PERTURBATIONS

We assess the robustness of LATTE under common post-processing operations like compression, resizing, Gaussian blur, and Gaussian noise. Such perturbations often occur in real-world pipelines and can severely degrade the subtle artifacts that detection methods depend on. As shown in Figure 5, LATTE consistently outperforms LaRE, maintaining higher detection accuracy and greater stability. This shows that LATTE's reliance on multi-step latent trajectories is more invariant under such transformations than single-step reconstruction errors.

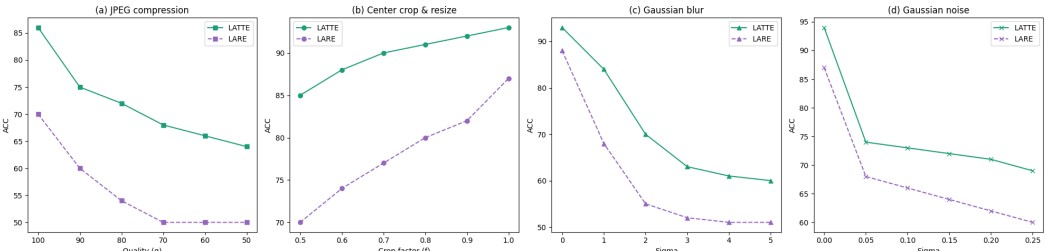

Figure 5: **Accuracy(%) of LATTE vs. LaRE on perturbed images.** We evaluate and compare the robustness of both methods under four common transformations: JPEG compression, center crop & resize, Gaussian blur, and noise. LATTE consistently outperforms LaRE across all perturbations.

## 4.6 QUALITATIVE ANALYSIS

We present qualitative examples in a confusion-matrix-style layout in Figure 6, highlighting representative model successes and failures. **Top-left**: Real images with complex textures, human subjects, or fine structures are typically recognized as authentic, since such details remain difficult for generative models to replicate. **Top-right:** In contrast, some real images with smooth textures, saturated colors, or stylized lighting are misclassified as fake, reflecting the model's sensitivity when authentic content visually resembles synthetic imagery. **Bottom-left:** On the other hand, high-quality generated images that appear simple or artifact-free may be mistaken for real, highlighting the difficulty of detecting visually convincing fakes. **Bottom-right:** Lastly, LATTE succeeds in correctly identifying visually convincing fake images, which suggests that it leverages subtle traces rather than only visual artifacts.

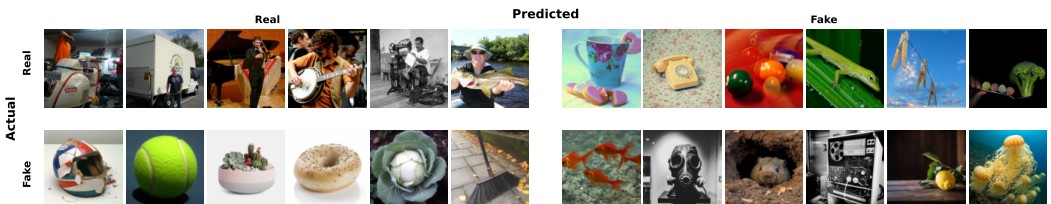

Figure 6: **Qualitative results in a confusion-matrix-style layout.** The rows show actual labels, and the columns show predictions of LATTE.

## 5 CONCLUSION

We propose **LATTE**, a novel diffusion-generated image detection approach that models the sequential evolution of latents across multiple denoising steps. By capturing trajectory patterns and grounding them with visual features, LATTE learns a compact and discriminative representation. Experiments on GenImage, Chameleon, and Diffusion Forensics demonstrate that LATTE achieves state-of-the-art performance, including significant gains in cross-generator and cross-domain scenarios. Overall, this work highlights latent trajectory modeling as a new direction for generated image detection.

**Limitations.** While LATTE achieves strong performance and improved generalization, its effectiveness diminishes under strong post-processing (e.g., heavy JPEG compression or strong blur), indicating sensitivity to distribution shifts. Additionally, like most global detectors, LATTE has been evaluated primarily on fully synthetic versus real images, while detecting small, localized forgeries remains a distinct challenge for future work.

ETHICS STATEMENT

This work advances the field of synthetic media forensics by improving the detection of generated images. As generative models improve their ability to produce highly realistic content, frameworks or tools like LATTE, play an important role in combating disinformation, verifying content authenticity, and maintaining public trust in digital media.

However, the deployment of the detection system also raises important ethical and societal considerations. As detection technologies improve, so do adversaries' strategies for evading them, potentially resulting in an arms race between generation and detection. Furthermore, there is a risk that such tools will be misapplied, for example, by incorrectly labeling legitimate content as false or by being employed in politically or socially biased ways. Overreliance on automated systems is another growing concern, as they may miss edge cases or fail silently in unfamiliar situations.

REPRODUCIBILITY STATEMENT

We are committed to ensuring the reproducibility of our results. To this end, we will release the full source code and evaluation scripts upon publication. Our paper clearly and fully describes the proposed feature extraction method and the model architecture in Section 3, and provides comprehensive details on the experimental setup in Section 4.1, including used datasets, preprocessing steps, training configurations, and hyperparameters. Additionally, ablations and variant evaluations in Section 4.3 and Appendix A further support reproducibility.

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

APPENDIX

The appendix consists of the following sections: A. Additional Ablation Studies, B. Latent Trajectory Spatial Analysis, C. Complete Accuracy and AP on GenImage, D. Architectural Details of the CLS-pooling, and E. Embedding Space Analysis.

## A  ADDITIONAL ABLATION STUDIES

To further understand the key design choices and components of the LATTE framework, we conduct a series of additional ablation studies. All ablation results reported in this section are based on models trained using the SDv1.4 subset of GenImage (Zhu et al., 2023b).

### A.1  BENEFIT OF AVERAGE POOLING

Standard pooling in LATTE assumes equal importance across all timesteps in the trajectory. To test this design choice, we experiment with a weighted pooling mechanism that assigns importance scores to each timestep using a linear gating function and softmax normalization. As shown in Table 7, this approach performs worse than simple average pooling - suggesting that all steps provide equally informative signals. We also consider CLS pooling, where a special token aggregates the sequence of latents through self-attention with positional encodings. The goal is to assess whether allowing the latents to refine each other via self-attention and incorporating sequence order can improve performance. This variant slightly underperforms, suggesting that LATTE is already expressive enough without additional attention-based aggregations.

Table 7: **Accuracy(%) comparison between different aggregation strategies.** Average pooling outperforms learnable weighted pooling by 9.4% and CLS pooling by 1.4%.

| Configuration | Midjourney | SDv1.4 | SDv1.5 | ADM | GLIDE | Wukong | VQDM | BigGAN | *Avg.* |
|---|---|---|---|---|---|---|---|---|---|
| Weighted pooling | 73.0 | 99.8 | 99.7 | 72.0 | 79.0 | 98.7 | 74.4 | 56.3 | 81.6 |
| CLS pooling | 85.8 | 99.0 | 99.0 | **83.2** | 86.7 | 96.8 | **88.7** | 77.8 | 89.6 |
| **Average pooling** | **88.8** | **100** | **99.9** | 74.0 | **95.8** | **98.9** | 80.8 | **89.8** | **91.0** |

### A.2  EFFECT OF LATENT EXTRACTION CONFIGURATION

The sequence of latents is obtained by first encoding real and fake images into latent space using a frozen VAE, followed by partial reconstruction via a pre-trained diffusion model. At each timestep, noise is added to the VAE latents and then partially denoised via the UNet, capturing intermediate latent representations along the reconstruction trajectory.

We ablate two factors in this latent extraction pipeline: the choice of sampling method (DDPM vs. DDIM, Table 8) and the choice of U-Net backbone (Stable Diffusion v1.5 vs. v2.1, Table 9). For the sampling method, we use Stable Diffusion v2.1 as the backbone, while for the U-Net model comparison, we fix the scheduler to DDPM.

Table 8: **Accuracy(%) comparison between DDPM and DDIM-based latent extraction.** DDPM improved accuracy by 7.2%.

| Sampling method | Midjourney | SDv1.4 | SDv1.5 | ADM | GLIDE | Wukong | VQDM | BigGAN | *Avg.* |
|---|---|---|---|---|---|---|---|---|---|
| DDIM | 77.0 | 99.7 | 99.5 | **74.6** | 82.2 | 98.2 | 77.4 | 61.8 | 83.8 |
| **DDPM** | **88.8** | **100** | **99.9** | 74.0 | **95.8** | **98.9** | **80.8** | **89.8** | **91.0** |

Table 9: **Accuracy(%) comparison between SDv1.5 and SDv2.1-based latent extraction.** SDv2.1 improves accuracy by 3.9%.

| U-Net Backbone | Midjourney | SDv1.4 | SDv1.5 | ADM | GLIDE | Wukong | VQDM | BigGAN | *Avg.* |
|---|---|---|---|---|---|---|---|---|---|
| SDv1.5 | 81.7 | 99.7 | 99.5 | **77.7** | 90.6 | 98.0 | 78.1 | 71.5 | 87.1 |
| **SDv2.1** | **88.8** | **100** | **99.9** | 74.0 | **95.8** | **98.9** | **80.8** | **89.8** | **91.0** |

The results indicate that LATTE's performance is sensitive to the latent extraction configuration, highlighting the importance of both the sampling method and the U-Net backbone. Switching from DDIM to DDPM yields a substantial improvement in average detection accuracy (+7.2%), with particularly large gains on datasets such as Midjourney and BigGAN. This suggests that the stochastic denoising dynamics captured by DDPM produce richer latent trajectories, enhancing the discriminative signal between real and generated images. Similarly, upgrading the U-Net backbone from SDv1.5 to SDv2.1 further improves average accuracy (+3.9%), reflecting the impact of more expressive latent representations on the model's ability to capture subtle generative artifacts. While some datasets, such as ADM, show minimal changes, likely due to inherent detection difficulty or saturation effects, the overall trend confirms that both the scheduler and backbone play complementary roles: the scheduler shapes the temporal evolution of latents, whereas the backbone determines the quality of the underlying feature space. Despite these variations, LATTE maintains high and consistent performance across all configurations, demonstrating its robustness and reliability as a diffusion-generated image detector.

A.3    INFLUENCE OF VISION BACKBONE FINE-TUNING

Our default setup fine-tunes the vision encoder. To quantify the added benefit of this choice, we compare against a variant where we freeze the backbone and train only the LATTE-specific components. Table 10 reports per-generator accuracy for both settings. We observe major improvements for both vision backbones when fine-tuned, with 15.5% accuracy gain for CLIP ViT-L/14 (Radford et al., 2021) and 9% for ConvNeXt (Liu et al., 2022). This likely stems from the fact that frozen backbones retain features that were never explicitly optimized for real vs. fake discrimination, leading to an embedding space that is misaligned with the objectives of generated image detection. Without adaptation, our model may struggle to effectively ground latent trajectories in meaningful visual semantics. Fine-tuning, by contrast, enables the backbone to specialize its representations for this task, enhancing the alignment between visual and latent features essential for robust detection.

Table 10: **Accuracy (%) comparison for different vision backbones and fine-tuning vs. frozen settings on the GenImage dataset.** Fine-tuned ConvNeXt yields the best performance.

| Backbone | Setting | Midjourney | SDv1.4 | SDv1.5 | ADM | GLIDE | Wukong | VQDM | BigGAN | *Avg.* |
|---|---|---|---|---|---|---|---|---|---|---|
| CLIP ViT-L/14 | Frozen | 60.3 | 99.8 | 99.9 | 53.6 | 50.3 | 99.8 | 51.4 | 50.1 | 70.6 |
| | Fine-tuned | **98.2** | 99.9 | **100** | 68.4 | 91.8 | **99.9** | 58.9 | 72.1 | 86.1 |
| ConvNeXt | Frozen | 79.7 | 99.3 | 99.1 | 64.4 | 74.2 | 95.9 | 72.4 | 70.9 | 81.9 |
| | Fine-tuned | 88.8 | **100** | 99.9 | **74.0** | **95.8** | 98.9 | **80.8** | **89.8** | **91.0** |

A.4    INFLUENCE OF SEPARATE LATENT PROCESSING STRATEGY

The default LATTE architecture, as described in Section 3, processes the latent trajectory by refining each timestep independently using a dedicated transformer decoder. An alternative approach is to stack the latent embeddings from all timesteps into a single sequence and process them jointly through a shared transformer decoder stack, enforcing full parameter sharing across the sequence. As shown in Table 11, decoding each timestep separately achieves higher accuracy across most generators, suggesting that preserving per-timestep decoding helps the model retain specific features from the denoising trajectory.

Table 11: **Accuracy(%) comparison** between separate vs. joint latent processing strategies. Processing timesteps separately yields the highest average accuracy, outperforming joint processing by 0.8%.

| Latent strategy | Midjourney | SDv1.4 | SDv1.5 | ADM | GLIDE | Wukong | VQDM | BigGAN | *Avg.* |
|---|---|---|---|---|---|---|---|---|---|
| Joint | 88.4 | 99.7 | 99.6 | 72.5 | 94.4 | 98.6 | 79.4 | 88.9 | 90.2 |
| **Separate** | **88.8** | **100** | **99.9** | **74.0** | **95.8** | **98.9** | **80.8** | **89.8** | **91.0** |

A.5    EFFECT OF POSITIONAL ENCODINGS IN CLS-POOLING

We conduct an ablation to isolate the effect of the positional embeddings when using CLS-pooling. Specifically, we compare the full model ("CLS-pooling w/ pos. enc.") to a variant that uses the same

CLS-based self-attention but omits positional embeddings ("CLS-pooling w/o pos. enc."), removing any explicit indication of timestep order. As shown in Table 12, providing sequence order information results in a significant improvement of 7.2% in average accuracy, confirming that timestep position is an important signal when aggregating latents jointly. Despite this gain, the CLS-based variant remains less effective than the default LATTE architecture, which aggregates the outputs via average pooling. Interestingly, the "CLS-pooling w/ pos. enc." variant demonstrates better performance on certain subsets - 9.2% increase on ADM and 7.9% on VQDM - suggesting that this CLS-based design, paired with sequence order cues, can be beneficial in specific contexts.

Table 12: **Accuracy(%) comparison for CLS-pooling with and without explicit sequence order.** Explicit positional embeddings improve accuracy by 7.2% over the implicit variant, but fall slightly short of the average pooling.

| Sequence order | Pos. enc. | Midjourney | SDv1.4 | SDv1.5 | ADM | GLIDE | Wukong | VQDM | BigGAN | *Avg.* |
|---|---|---|---|---|---|---|---|---|---|---|
| CLS-pooling | no | 75.5 | 99.7 | 99.6 | 72.2 | 78.9 | 98.3 | 75.6 | 59.3 | 82.4 |
| CLS-pooling | yes | 85.8 | 99.0 | 99.0 | **83.2** | 86.7 | 96.8 | **88.7** | 77.8 | 89.6 |
| **Avg. pooling** | N/A | **88.8** | **100** | **99.9** | 74.0 | **95.8** | **98.9** | 80.8 | **89.8** | **91.0** |

## B  LATENT TRAJECTORY SPATIAL ANALYSIS

To motivate the modeling of the latent trajectory and to distinguish how diffusion-based reconstructions differ between real and generated images, we analyze the spatial distribution of latent denoising corrections across timesteps.

Specifically, we compute the average per-pixel norm of latent differences between consecutive denoising steps - denoted as $\Delta z_t = |z_t - z_{t-1}|_2$ - for a sequence of tracked timesteps $t_1, t_2, \ldots, t_K$. For each timestep interval $t_{k-1} \to t_k$, we aggregate $\Delta z_t$ across all spatial positions and across a batch of samples to obtain a mean spatial correction heatmap:

$$H_{t_k}(x, y) = \mathbb{E}_n \left[ \left| z_{t_k}^{(n)}(x, y) - z_{t_{k-1}}^{(n)}(x, y) \right|_2 \right],$$

where $(x, y)$ indexes spatial coordinates and $n$ indexes the samples. The resulting heatmaps visualize how the latent representation evolves across timesteps by capturing the spatial magnitude of change between consecutive steps. They serve as a proxy for identifying where and how strongly the model updates its latent estimated at each stage of the denoising process. This spatial perspective complements our temporal trajectory modeling and helps reveal structural patterns that distinguish real and generated images.

Based on Figure 7, we observe a clear dichotomy between real and fake images across most generators. The real images follow a smooth, uniformly paced denoising trajectory, indicating that each denoising correction is modest in magnitude and spatially consistent.

Fake images, in contrast, break this steady pattern in different ways. Images generated by GLIDE (7a) require substantially larger corrections overall. The early steps are especially bright - indicating heavier refinement in the beginning of the denoising process - before tapering off into smaller updates. Midjourney (7d) and BigGAN (7e) behave almost identically, with lower differences between real and fake heatmaps than Glide, but still pronounced at every step. Unlike the real's constant gradual decline, their fake trajectories show a striking front-loaded burst: the jump in $\Delta z$ between the first two steps is far greater than any subsequent change. This pattern reveals that, for these generators, most of the refinement occurs in the first half of the trajectory, with little correction applied later.

By contrast, the ADM subset (7b) shows a markedly different trend. Here, the real vs. fake differences across all steps are considerably more subtle, and the resulting $\Delta z$ heatmaps for both classes appear visually similar in both magnitude and spatial pattern, with the exception of small brighter left and top margins. This behavior is consistent with our model's relatively poor performance on ADM (74% compared to the 91% average) and suggests that the images in this subset are particularly difficult to distinguish - even in trajectory space.

Finally, SDv1.4 (7c) presents the most distinctive behavior. Unlike previous generators, the fake heatmaps exhibit a center-focused $\Delta z$ signature. This effect likely arises because we use Stable

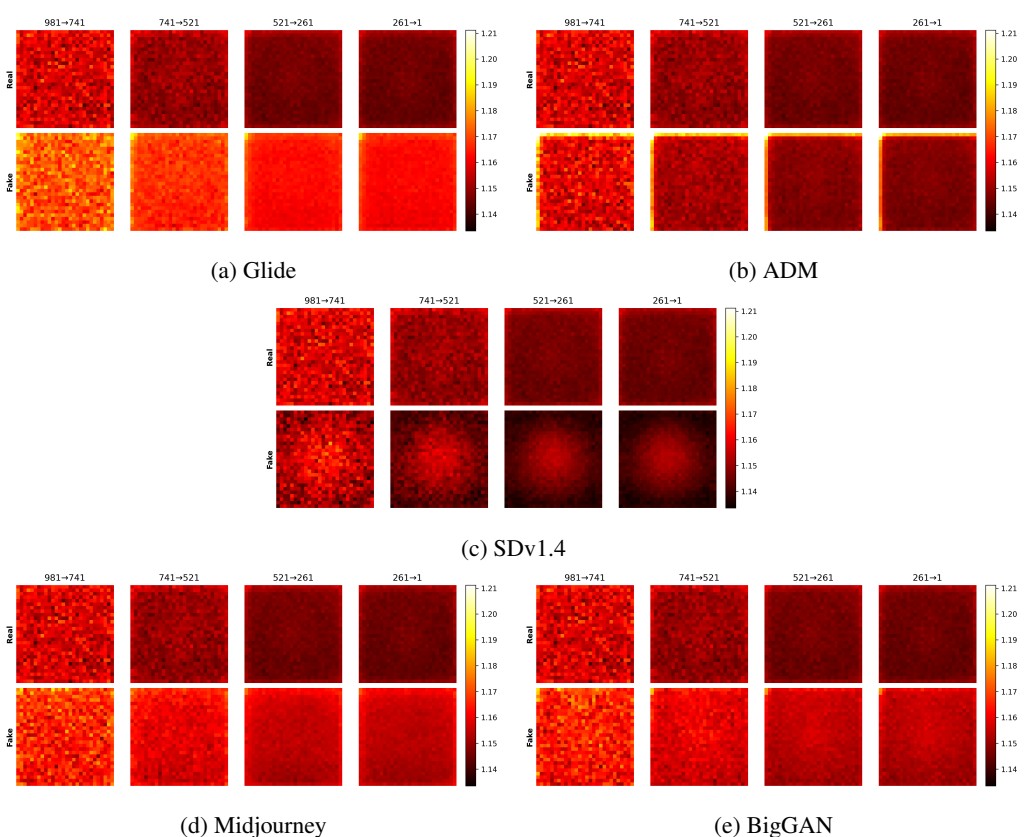

Figure 7: **Latent trajectory spatial analysis using images from the GenImage dataset.** The real images plots represent averages over all real images in the dataset, while the fakes are plotted separately based on the generators used to produce them.

Diffusion (Rombach et al., 2022b) for both generating and reconstructing the images. The denoiser has learned to prioritize central content - where objects are typically located during prompt-guided generation - and thus applies larger, spatially focused corrections in the center of the image. Real images, by contrast, lack this learned structure and receive relatively uniform and lower-magnitude corrections across space.

## C COMPLETE ACCURACY & AVERAGE PRECISION ON GENIMAGE

Figure 8 presents LATTE's performance on the GenImage dataset, reporting both accuracy and average precision across different training–testing generator combinations. The results show that LATTE maintains consistently high performance regardless of the generator used for training, highlighting its ability to generalize across diverse generative models.

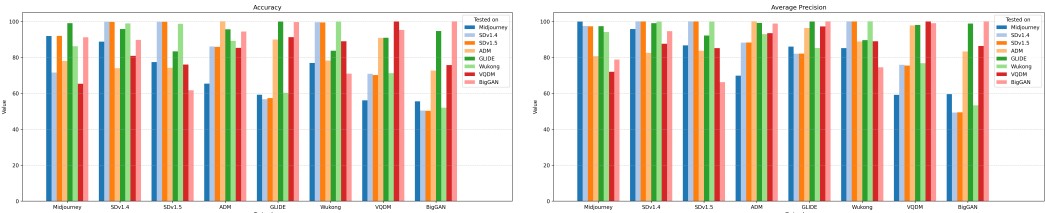

Figure 8: **Accuracy(%) (left) and average precision(%) (right) of LATTE across the GenImage dataset.** The x-axis indicates the generator used to produce the training data, while each bar represents the model's performance when tested on data from the different generators.

## D ARCHITECTURAL DETAILS OF THE CLS-POOLING

We consider CLS-pooling as an alternative aggregation strategy (instead of average pooling), illustrated in Figure 9. After independently fusing each projected latent $\tilde{z}_t$ with visual features, through a stack of transformer decoder layers, a learnable token $z_{\text{CLS}} \in \mathbb{R}^d$ is prepended to the sequence of refined latent embeddings $\widetilde{\mathcal{T}}(x) = \{\tilde{z}_{t_1}, \tilde{z}_2, \ldots, \tilde{z}_{t_K}\}$. Learnable positional embeddings are added to this sequence to inform the model of the order of timesteps. The sequence is then passed through a shared self-attention stack of transformer layers, allowing the CLS token $z_{\text{CLS}}$ to interact with the full latent trajectory and aggregate information across timesteps. The final CLS token output serves as the aggregated trajectory representation $\tilde{z}_{\text{agg}}$. The rest of the architecture remains the same as in Figure 2.

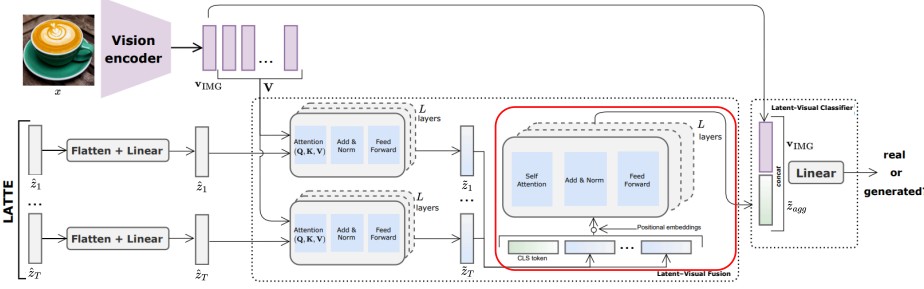

Figure 9: **Overview of our proposed LATTE architecture with CLS pooling as an aggregation strategy (denoted in red).** A learnable CLS token is prepended to the fused sequence and processed via a self-attention stack.

## E EMBEDDING SPACE ANALYSIS

To complete our embedding space analysis from Section 4.4, Figure 10 presents t-SNE plots for the three remaining subsets in the GenImage dataset, namely the SDv1.5 (Rombach et al., 2022b),

Wukong (Wukong, 2024), and VQDM (Gu et al., 2022) generators. As in Figure 4, the top row shows embeddings extracted with the frozen ConvNeXt backbone (Liu et al., 2022) (pre-LATTE) and the bottom row shows embeddings after LATTE fine-tuning. The much clearer separation in the second row illustrates LATTE's discriminative power.

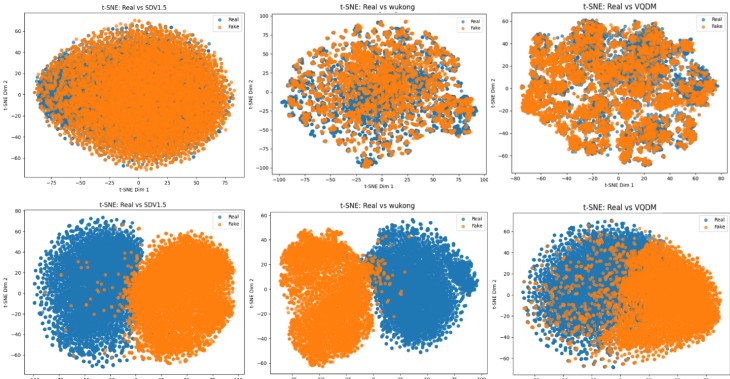

Figure 10: **Visualizations of t-SNE embeddings for real and fake images across the remaining three generators from GenImage.** The first row presents embeddings before using LATTE (extracted using the original ConvNeXt), while the second row shows embeddings derived from LATTE.

## LLM USAGE

Large language models, such as GPT-5 (OpenAI, 2025), were used only for manuscript preparation, including text polishing and grammar correction. All scientific contributions, formulating the research ideas, designing the methodology, conducting the experiments, and collecting results were conceived, developed, and validated by the authors.

