# OpenReview forum: "LATTE: Latent Trajectory Embedding for Diffusion-Generated Image Detection"
_ICLR.cc/2026/Conference — ICLR 2026 Conference Withdrawn Submission_

### Official Review · Reviewer_2rqs · 2025-10-25

**Soundness:** 2
**Presentation:** 2
**Contribution:** 2
**Rating:** 4
**Confidence:** 4

**Summary:**

The authors propose LATent Trajectory Embeddings (LATTE) to tackle the problem of synthetic image detection. Specifically, they claim that their method is the first to leverage diffusion trajectories in this context. The method is benchmarked on GenImage, Chameleon and Diffusion Forensics datasets where the SOTA performance is claimed.

**Strengths:**

1. The paper is easy to read and the overall methodology is clear.
2. The evaluation is thorough and includes several baselines, cross-domain comparison and a robustness analysis to common corruptions.
3. The components of the proposed method are ablated such that the contribution of different features is understood.

**Weaknesses:**

1. The idea of using information in trajectories is not new. Consider including a discussion on the connection of this work with prior findings. For example, departure from single-step errors to exploit the multi-step nature of diffusion has been previously established in [1].

2. The authors justification for the inclusion of LATTE in the overall pipeline is unclear. The ablations in Table 4 show that the LATTE features alone are not effective for the task. Instead, it is the visual backbone features that are most informative.

3. Beyond improved metrics, there is little insight regarding the diffusion trajectories and how they may encode discriminative signals. The discussion and experiments here should be improved, especially since modeling trajectory evolution is advertised as one of the main contributions of the paper. The analysis in Appendix B is a good first step toward this but it is not entirely clear (e.g., see Questions 1 and 4 below) and the authors do not convincingly present arguments that motivate the trajectory analysis.

**Questions:**

1. The authors model the LATTE features with five timesteps that are evenly spread across the trajectory. Is such an allocation optimal or are there regions of the trajectory that are more informative?

2. It seems that most of the results emphasize performance on synthetic data while performance on real samples is omitted (e.g., Table 1, Figure 3). Could you please update your benchmarks to reflect overall performance? It is not possible to judge a binary classifier's performance based on conditional metrics.

3. The authors have performed extensive experiments to find the best visual backbones and diffusion backbones. Given the thorough ablations, I am skeptical regarding methodological effectiveness: To what extent is the performance of your method attributed to the pretraining of the backbones vs. LATTE?

4. Line 54: "We hypothesize that real images, whose details and textures can lie outside the model’s learned manifold, will often produce small inconsistencies between successive denoising steps. On the contrary, fake images will follow smoother, more self-consistent trajectories aligned with the model’s generative prior". However, Appendix B (line 900) states: "Based on Figure 7, we observe a clear dichotomy between real and fake images across most generators. The real images follow a smooth, uniformly paced denoising trajectory, indicating that each denoising correction is modest in magnitude and spatially consistent."?

[Minor] The paper is on synthetic image detection, consider rewording the title. For example, you also benchmark on samples from BigGAN, which I assume are ideally classified as generated in your framework.

[Minor] Equation 6 does not correspond to reverse diffusion (see Algorithm 2 in DDPM paper).

[1] Tracing the Roots: Leveraging Temporal Dynamics in Diffusion Trajectories for Origin Attribution, 2024

---

### Official Review · Reviewer_ajtf · 2025-10-28

**Soundness:** 3
**Presentation:** 3
**Contribution:** 2
**Rating:** 2
**Confidence:** 5

**Summary:**

The authors propose detecting a new method for fake image detection. The core idea is an extension of previous works which suggested that a real and fake image’s reconstruction (by the encoder + decoder) is different - the reconstruction is more accurate for fake images than for real images, a signal that can be used to detect fake images. The authors argue that simply considering the single step reconstruction misses out on other information available in the overall denoising process. Consequently, for a test image, they perform a complete forward and backward (denoising) process, and obtain the intermediate latents. These latents from different time steps are fused with visual features of the same image using attention mechanisms. Finally, the real-vs-fake detection is performed on a combination of visual and latent features. Authors demonstrate improved performance compared to baselines on multiple benchmarks.

**Strengths:**

- The authors’ key idea is that because of how diffusion models work, we do not need to limit ourselves to learning a classifier solely using the image pixels of real and fake images. Instead, we can obtain their intermediate latents, which lead up to the final image, and see how all those latents differ for real and fake images. This is a very reasonable idea, and, to the best of my knowledge, has not been explored before.

- The authors have evaluated their method on multiple datasets and showed improvement in cross-generator generalization compared to multiple baselines. Especially the results on two benchmarks - GenImage (Table 1) and
Diffusion Forensics (Table 3) - show that the improvements over the baselines are significant.

- The paper is simply written and easy to understand.

**Weaknesses:**

- While the idea of using latents of real and fake images makes sense, there are many other areas where the paper lacks strong motivation and analysis. One of these is the idea of mixing the latents with the visual image features. I will list a couple of sections of the paper which are somewhat vague and do not provide a concrete explanation about this process.

    - Line 194 - 195: Why is it important to ground the latent features to the visual context through the attention mechanism? In other words, what is the conceptual issue (which is independent from empirical results) with not performing such grounding?

    - Line 206 - what does it mean for the latents to be “enhanced”?

    - Line 236 - What does it mean for the latents to be “enriched”?

- Overall, the rationale for fusing these two modalities of data is not grounded sufficiently in some fundamental principle, and certain words that the authors have used (e.g., enhanced, enriched) seem pretty vague.

- Furthermore, the analysis of the results is lacking. A few examples.

    - Lines 362-363 (“As shown in Table 4 …”) : this does not really explain why this is to be expected. It ties to the weakness mentioned above - it is not clear why the fusion is needed. Even beyond the fusion, it is not clear what additional information the latents are providing. I have already mentioned in the strengths that the usage of latents seems to be a new idea, but it is not clear what kind of information does it present the final linear classifier. Do the latents better capture the low level artifacts, or high level artifacts?

    - Lines 400-401: no explanation is given for why these backbones perform better.

    - It is not clear why the proposed method should be able to detect StyleGAN images (Table 3), even though that is not the main point of the proposed algorithm, since it cares only about the latents from a diffusion model?

- While it will certainly be good if the authors can answer some of these questions, my objective is to point out that, right now, the paper is lacking substantial analysis of (i) the approach and (ii) results, and reads more like an idea which empirically seems to give good results. Consequently, it is not clear what the key insight the reader is supposed to get out of this work.

- The proposed method is going to be computationally more expensive since it requires for each real and fake image to extract their latents at multiple time steps and also to fuse them with the visual features. But no discussion of the additional time complexity is presented.

- The authors have not explained what real images are used during training.

**Questions:**

- Figure 2, what parts are trained? Is the pretrained vision encoder also trained (finetuned)?

- Line 390: the indices associated with n=5 seem a little bit arbitrary. Is there a reason why any sequence of steps at 200 steps away (e.g., 1, 201, 401, 601, 801) won’t work?

---

### Official Review · Reviewer_JFdm · 2025-10-29

**Soundness:** 2
**Presentation:** 3
**Contribution:** 2
**Rating:** 2
**Confidence:** 5

**Summary:**

LATTE: LATENT TRAJECTORY EMBEDDING FOR DIFFUSION-GENERATED IMAGE DETECTION

The paper focuses on a timely and important problem, i.e., how to detect the diffusion-generated images. Specifically, it proposes an approach that models the evolution of latent embeddings across multiple denoising steps. Comparing against former methods, it achieves SOTA.

**Strengths:**

The paper focuses on a timely and important topic.

The paper is generally well-written and easy to follow.

The images are clear .

**Weaknesses:**

The paper has the following concerns:

1. Given the nature of the proposed method, the inference speed should be very slow, which limits the real-world deployment. And there is no experiments focusing on efficiency.
2. The paper only focuses on diffusion-generated images. However, the SOTA generation models, such as GPT-Image-1 and Nano Banana maybe auto-regression based.
3. No latest model like FLUX is discussed.
4. The novelty of the proposed method is below the bar of ICLR. For myself, Latent–Visual Fusion and Latent-Visual Classifier are not novel enough for top-tier publications.

**Questions:**

I suggest rejection. Nothing can change my mind.

---

### Official Review · Reviewer_9Gfv · 2025-10-31

**Soundness:** 1
**Presentation:** 2
**Contribution:** 2
**Rating:** 2
**Confidence:** 5

**Summary:**

This work aims to build a robust fake image detector. The authors propose LATTE, a type of reconstruction-based detection. The key hypothesis is that the whole trajectory of the multi-step diffusion process has valuable information which can be leveraged in order to detect fake images. Based on this, the denoised latent conditioned on various timesteps are concatenated and used as a signal to train a fake image detector on. The LATTE method outperforms several popular baselines on established benchmarks.

**Strengths:**

1. Achieves good results on several benchmarks outperforming several prominent baselines.

**Weaknesses:**

Weaknesses
1. Equations 5/6, rely on the randomly sampled noise (epsilon). At higher noise levels, this would cause huge differences in the outputs produced by the network, hence there is a chance that this could affect the features extracted by the neural network. To clarify these doubts, the authors should present all the results as a mean/standard deviation over 10-15 evaluations. This sheds doubts upon one result (in Fig 3, training on SDv1.4 generalizes to BigGAN, but training on SDv1.5 does not). However, both of these are very similar distributions (detectors which train on one of these work on the other one).
2. Related to above, in the Appendix A.2, why does DDIM perform so much worse than DDPM. The provided explanation of DDPM trajectories being "richer" than DDIM does not seem intuitive. Further explanation + additional evaluations as stated above would help alleviate these doubts.
3. Experiment studying the importance of denoising steps, does not provide convincing evidence for the need for the whole trajectory information (which is the central claim). In Table 5 (n=1) which does not select the first timestep (most signal) achieves the lowest performance. This could suggest that in fact only the initial timesteps are useful and not the whole trajectory information. Further doubt is cast by the fact that if more 9 steps are sampled, the performance is severely affected. This seems contrary to the original hypothesis which emphasizes that extracting more information from the trajectory should help with detection.
4. Perturbation analysis (Fig 5) should also compare with other methods (not only LARE). It is concerning that the performance drops for JPEG 50, where the image does not look overly distorted. It would be interesting to know if this method can be trained with some common post-processing operations as augmentations (would help with robustness).
5. It would be beneficial to report AP values of the baselines as well so that the full potential of the detector can be analyzed. Currently, the paper only reports the AP of the LATTE method on GenImage.

**Questions:**

1. Related to weaknesses, but lot of doubts on what the average and standard deviations are for each of the readings, as well as the effect actual influence of the whole trajectory information (given the information in Table 5).

---

### Note · Authors · 2025-11-12

I have read and agree with the venue's withdrawal policy on behalf of myself and my co-authors.